# RNA-Seq Study on the *Longissimus thoracis* Muscle of Italian Large White Pigs Fed Extruded Linseed with or without Antioxidants and Polyphenols

**DOI:** 10.3390/ani13071187

**Published:** 2023-03-28

**Authors:** Jacopo Vegni, Ying Sun, Stefan E. Seemann, Martina Zappaterra, Roberta Davoli, Stefania Dall’Olio, Jan Gorodkin, Paolo Zambonelli

**Affiliations:** 1Department of Agricultural and Food Sciences (DISTAL), University of Bologna, 40127 Bologna, Italy; 2Center for Non-Coding RNA in Technology and Health, Department of Veterinary and Animal Sciences, University of Copenhagen, 1870 Copenhagen, Denmark

**Keywords:** swine, transcriptome, methodological approach for RNA-seq data analysis, polyunsaturated fatty acid (PUFA), antioxidant, vitamin E, selenium, polyphenols

## Abstract

**Simple Summary:**

In humans, a dietary intake of omega-3 polyunsaturated fatty acids along with antioxidants has been shown to have anti-inflammatory and antioxidant activities. In pigs, on the other hand, there are few studies dealing with the use of omega-3 polyunsaturated fatty acids in the diet. For this reason, our study aimed to investigate the differences in gene expression of the Longissimus thoracis muscle of Italian Large White pigs fed with four different diets: a standard diet for growing-finishing pigs and three experimental diets; one supplemented with extruded linseed, a source of omega-3 polyunsaturated fatty acids, another with extruded linseed plus vitamin E and selenium as antioxidants, and another with extruded linseed plus oregano and grape skin extracts, which are natural polyphenols. From the results of the expression analysis, it was possible to deduce that, in the diets, the oxidative stability of the *n*-3 fatty acids increased, consistent with an increase in the fluidity of cell membranes, and increasing the anti-inflammatory potential of muscle. This can determine the high quality of the muscle tissue as regards the lipid composition; consequently, the meat will be qualitatively better for human health.

**Abstract:**

The addition of *n*-3 polyunsaturated fatty acids (*n*-3 PUFAs) to the swine diet increases their content in muscle cells, and the additional supplementation of antioxidants promotes their oxidative stability. However, to date, the functionality of these components within muscle tissue is not well understood. Using a published RNA-seq dataset and a selective workflow, the study aimed to find the differences in gene expression and investigate how differentially expressed genes (DEGs) were implicated in the cellular composition and metabolism of muscle tissue of 48 Italian Large White pigs under different dietary conditions. A functional enrichment analysis of DEGs, using Cytoscape, revealed that the diet enriched with extruded linseed and supplemented with vitamin E and selenium promoted a more rapid and massive immune system response because the overall function of muscle tissue was improved, while those enriched with extruded linseed and supplemented with grape skin and oregano extracts promoted the presence and oxidative stability of *n*-3 PUFAs, increasing the anti-inflammatory potential of the muscular tissue.

## 1. Introduction

To date, there are multiple strategies used to improve the nutritional quality of meat and meat products [1]. These include adding sources of *n*-3 polyunsaturated fatty acids (*n*-3 PUFAs) and antioxidants, such as selenium plus vitamin E or natural polyphenols, to the diet. In humans, *n*-3 PUFAs co-added with antioxidants have a positive role in the metabolism by showing anti-inflammatory and antioxidant activity and have a positive effect against obesity and insulin resistance [2]. In swine, few studies in the literature examined the effects on metabolism, particularly at the molecular level, of dietary intake of *n*-3 PUFAs sources supplemented with antioxidants and polyphenols.

This research aims to study pig *Longissimus thoracis* muscle gene expression differences between pig diets through the application of a selective workflow of RNA-seq data processing. Compared with previous studies, we chose DESeq2 to identify the differential expression genes (DEGs) and we applied a strict log2 Fold Change (log2FC) to identify the DEGs. This approach, which is quite common in human research, was used in pigs to identify effects on gene expression of diets supplemented with different antioxidants. Thanks to the identification of differentially expressed genes, this paper highlights the relevance of adding antioxidants to pig diets when animals are fed with a source of polyunsaturated fatty acids, in order to increase the stability of the fat component of pork utilized both for fresh consumption and to produce high-quality pig-meat-seasoned products.

## 2. Materials and Methods

Forty-eight Italian Large White pigs, 24 gilts and 24 barrows, were used in the experiment. These pigs were chosen from a large group of 258 piglets, which were descended from 21 sows and 3 boars marked in the herd book of the Italian National Association of Pig Breeders (ANAS; [3]).

The 48 pigs were divided into four experimental groups of 12 animals, each balanced for weight, father, and sex. The subjects were all fed a standard diet until the start of the trial, after which each group was given its respective diet, which was a standard diet for growing-finishing pigs (D1); the same diet as D1, enriched with extruded linseed, an *n*-3 PUFAs source (D2); the same diet as D2, enriched with vitamin E and selenium (D3); and the same diet as D2, enriched with grape skin and oregano extracts, sources of natural polyphenols (D4). In the middle of the trial, in the experimental group D4, a pig died of natural causes. For ingredients, chemical composition, and feeding methods of the four diets administered, for the manner and timing (in relation to the weight of the animals) of pig slaughtering, and for regulations on the protection of animals at slaughter, refer to [2,4]. After slaughter, *Longissimus thoracis* muscle samples were taken and placed immediately into liquid nitrogen for cryopreservation. After that, they were stored at −80 °C until the time of RNA extraction. Regarding materials and methods of RNA extraction, library preparation, and sequencing, we refer to [2,4].

The forty-eight RNA-seq datasets for pigs fed with different diets were downloaded from the ArrayExpress [5] (accession: E-MTAB-7131), whose reads are 100 nucleotides paired-end sequencing reads. The quality of the raw reads was evaluated using FastQC v.0.11.5 [6] and reported in detailed files with MultiQC v.1.10.1 [7]. Then, the reads were trimmed with Trimmomatic v.0.39 [8,9] by removing Illumina adapters, deleting the final bases of the reads with quality <3, eliminating reads when their average quality was <15 in a sliding window of 4 bases, and, finally, removing reads of length <60 nucleotides to ensure the highest quality of clean reads. Following this, clean reads were mapped to the reference genome, *Sus scrofa* genome assembly version Sscrofa11.1 [10] using STAR v.2.6.1.d [11,12] with default parameters and uniquely mapped reads obtained after filtering were used for the quantification of gene expression. FeatureCounts was used for the evaluation of gene expression, implemented in Subread v.1.6.3 [13] using the default parameters, and based on the genomic annotation of swine (release-104) from Ensembl database [14]. The identified genes were then assessed for differential expression between experimental diets: D1 vs. D2, D1 vs. D3, D1 vs. D4, D2 vs. D3, D2 vs. D4, and D3 vs. D4, for a total of six comparisons.

DEGs were then detected using DESeq2 [15], an R package from Bioconductor v.3.14. In DESeq2, the correction method used anticipated the dietary groups as experimental factors, while father, sex, slaughter day, and hidden batch effect were fixed factors. The hidden batch effect was previously calculated with sva [16], an R package from Bioconductor v.3.14, to adjust for unknown, unmodeled, or latent sources of noise; noise that would have conditioned the effect exerted by diets [17]. Genes were assumed to be differentially expressed only in those with at least 8 samples in at least one condition, with a number of reads equal to at least 10. The same conditions were used in studies concerning humans [18]. In addition, DEGs were considered those fulfilling the criteria of log2FC ≥ |0.70| [19] and False Discovery Rate (FDR) adjusted *p*-value ≤ 0.1, preserving the highly expressed DEGs, and they detected and described, in particular, the most pronounced differences in gene expression between diets provided to pigs of the same breed. For the validation methodology with quantitative real-time PCR, refer to [2,4].

In order to perform a functional analysis, DEGs were considered. For the annotation of the DEGs, the pig annotation gene was used first, after which the remaining unidentified genes were named using the human homologous genes. To do this, BioMart-Ensembl [20] was employed [14]. The functional enrichment analysis of DEGs was analyzed using stringApp, an app of the Cytoscape v3.9.1 software [21], using databases Gene Ontology (GO, including Biological Process, Cellular Component, Molecular Function), KEGG Pathways, and Reactome Pathways. All the genes from *Homo sapiens* were used as the background for the analysis because, with this background, we obtained networks with more genes involved and more interactions than using *Sus scrofa*. For the realization of the network, a confidence (score) cutoff of 0.40 was used, and to favor the creation of a network that included genes relevant for functional analysis, but which were not present among the DEGs, 5 maximum additional interactor genes were added for the comparison of D1 vs. D3. No genes were summed for the comparison of D1 vs. D4, while for the comparison between D2 vs. D4, no network was built because of the small number of DEGs found therein. The functions and pathways considered in the study had a significance threshold of FDR < 0.05.

## 3. Results

The results of the RNA-seq data pre-processing and gene expression analysis are shown in Table 1. Not all clean reads were assigned to that feature, and this is probably because the pig genome was not completely annotated, so a part of the remaining reads was not assigned. However, through these reads, it would be possible to update the annotation of the pig genome [22].

As the result of differential expression assessment, a total of 36 significant DEGs were detected, of which 34 genes were unique and non-redundant considering all comparisons between diets. Only two DEGs (transmembrane protein with EGF-like and two follistatin-like domains 2, *TMEFF2*, and RING1 and YY1 binding protein, *RYBP*) were detected for D2 vs. D4, and thus were not further considered in functional analyses. For D1 vs. D3, 22 DEGs were obtained, of which 19 were upregulated in D3. Finally, for D1 vs. D4, 12 DEGs were obtained, of which 11 were upregulated in D4. Since we did not detect significant DEGs in the comparison of D1 vs. D2, D2 vs. D3, and D3 vs. D4, these were omitted from Table 2 and Table 3. The complete list of DEGs with their average expressions and significance is reported in Table 2. The list of pathways and functions detected by functional enrichment analysis is reported in Table 3, and the description of the roles of the DEGs considered in the study is shown in Table 4.

From the subsequent functional analyses of the D1 vs. D3, some pathways were detected using Cytoscape (Figure 1a). Among them, the “positive regulation of the immune system process” Biological Process of the GO database (Table 3) included the DEG lymphatic vessel endothelial hyaluronan receptor 1 (*LYVE1*), which is connected to the C-C motif chemokine ligand 21 (*CCL21*) and plasmalemma vesicle-associated protein (*PLVAP*) in the network (Figure 1a), and has a role in regulating immune cell migration (Table 4).

## 4. Discussion

The results obtained from the D1 vs. D3 comparison are consistent with the hypothesis that the cell migration process of the immune system is more activated in the diet supplemented with extruded linseed plus selenium and vitamin E (*CCL21*, [24]; *LYVE1*, [25,26]), and the filtering efficiency of lymphocytes within the blood vessels is stimulated (*C4A*, [23]; *PLVAP*, [27]). This may suggest that, in pigs, the intake of a diet enriched with *n*-3 PUFAs (extruded linseed) plus antioxidants (vitamin E and selenium) promotes a more rapid and massive immune system response because the overall function of muscle tissue is improved.

Considering the comparison of D1 vs. D4, the “Unsaturated fatty acid biosynthetic process” Biological Process of the GO database was detected as significant (Table 3) and included stearoyl-CoA desaturase (*SCD*), ELOVL fatty acid elongase 5 (*ELOVL5*), and ELOVL fatty acid elongase 6 (*ELOVL6*) genes (Figure 1b); the *ELOVL5* gene codes for an enzyme acting in the metabolic path of docosahexaenoic (DHA) and eicosapentaenoic (EPA) *n*-3 acid formation [29] from the alpha-linoleic supplementation [28]. This is consistent with the D4 supplementation stimulated the expression of genes acting in the synthesis of eicosanoid acids. In addition, dietary D4 supplements, grape skin, and oregano, with their antioxidant effects on lipids, might preserve EPA and DHA [32,33,34,35]. This could result in a greater concentration in the phospholipid membrane of these *n*-3 fatty acids—which are the precursors of important anti-inflammatory metabolites released upon inflammation, such as resolvins, protectins, and marensins—in muscle cells [36]. However, the lack of phenotypes limits the full understanding of how DEGs and signaling pathways influence certain characteristic meat traits.

## 5. Conclusions

To summarize, we identified 34 DEGs of which 27 are new DEGs compared to the 289 DEGs in [4]. Given the source of RNA being comparable pigs under different diets, we do not expect large changes in their transcriptional landscape (reflected by the low log2FC cut-off). Hence, to retrieve a set of DEGs with a lower number of false positives we conducted the present data analysis using more conservative filters and the statistical tool DESeq2 which was shown in [37] as preferential for a moderate number of replicates to call small numbers of true positive DEGs. The current data analysis suggests that the use of antioxidants (selenium and vitamin E) or polyphenols as natural antioxidants (grape skin and oregano) in the diets enriched with *n*-3 PUFAs derived from extruded linseed increased both the content and oxidative stability of *n*-3 fatty acids. This possibly provides the cells with greater membrane fluidity and anti-inflammatory potential, important requirements for maintaining cellular physiology as reported for immune cells [28], and allows for a higher quality of muscle tissue resulting in increased meat quality for human health in relation to the lipid content and composition [38]. In general, this paper highlights the relevance of adding natural antioxidants to pig diets when animals are fed with a source of polyunsaturated fatty acids in order to increase the stability of the fat component of pork produced by heavy pigs, which can be utilized both for fresh consumption and to produce high-quality pig-meat-seasoned products.

## Figures and Tables

**Figure 1 animals-13-01187-f001:**
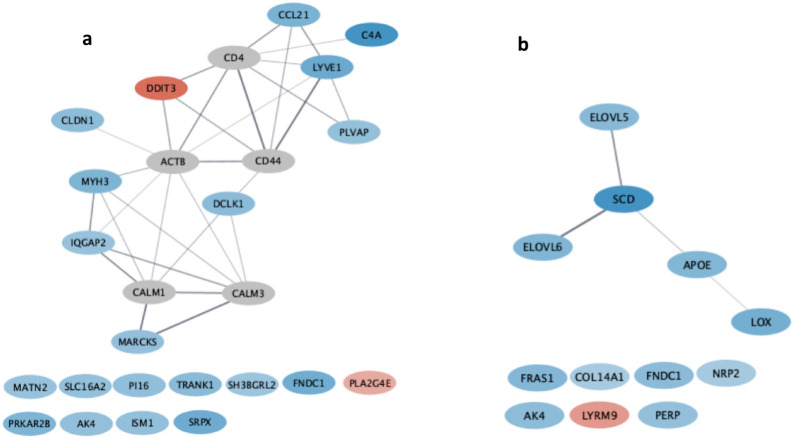
Cytoscape Gene networks achieved with stringApp for the DEGs obtained comparing D1 vs. D3 diets on the left (**a**) and D1 vs. D4 diets on the right (**b**). Over-expressed genes were colored in shades of blue, from the lightest (least over-expressed) to the darkest (most over-expressed). Down-expressed genes were colored in shades of red, from the lightest, least under-expressed to the darkest, most under-expressed. The genes indicated with a gray ellipse are additional interactor genes selected by Cytoscape.

**Table 1 animals-13-01187-t001:** Results of RNA-seq data pre-processing and gene expression analysis. The table shows, for each sample, the number of starting reads (raw reads), the number of reads remaining after the trimming step (clean reads), the percentage of reads out of the total clean reads uniquely mapped to the genome (uniquely mapped reads), and the number and percentage of reads assigned to exons out of the total clean reads for the identification of differential expression genes.

Sample	Raw Reads (*N*)	Clean Reads (*N*)	Uniquely Mapped Reads (%)	Reads Assigned (*N*)	Reads Assigned (%)
ERR2775643	15,222,091	12,231,608	86.9	9,139,962	74.7
ERR2775644	11,864,858	9,324,171	87.3	6,775,878	72.7
ERR2775645	19,689,337	15,667,592	85.9	11,375,307	72.6
ERR2775646	20,182,659	16,184,879	86.2	11,585,338	71.6
ERR2775647	12,477,450	9,885,484	86.2	7,077,650	71.6
ERR2775648	12,974,371	9,985,754	87.3	7,270,811	72.8
ERR2775649	12,534,078	9,874,602	85.9	7,131,936	72.2
ERR2775650	13,908,762	10,890,036	85.9	7,959,151	73.1
ERR2775651	15,976,939	12,860,711	86.5	9,412,286	73.2
ERR2775652	13,209,059	10,380,751	87.8	7,545,142	72.7
ERR2775653	16,449,679	13,227,601	85.9	9,420,488	71.2
ERR2775654	17,059,831	13,680,926	85.9	9,809,351	71.7
ERR2775655	17,665,490	14,373,661	86.2	10,242,721	71.3
ERR2775656	19,899,598	16,125,093	85.9	11,299,711	70.1
ERR2775657	15,916,725	12,643,514	86.1	8,989,816	71.1
ERR2775658	11,677,680	9,152,447	85.9	6,543,642	71.5
ERR2775659	15,434,949	12,447,467	86.4	9,135,322	73.4
ERR2775660	17,757,331	14,217,671	85.5	10,410,825	73.2
ERR2775661	13,390,239	10,536,701	87.0	7,571,975	71.9
ERR2775662	16,184,027	12,817,542	85.7	9,264,611	72.3
ERR2775663	14,471,427	11,460,053	86.8	8,240,029	71.9
ERR2775664	17,320,618	13,933,143	85.7	10,027,479	72.0
ERR2775665	11,995,565	9,496,805	85.5	6,876,554	72.4
ERR2775666	17,203,151	13,983,413	86.9	10,233,830	73.2
ERR2775667	14,287,420	11,591,105	86.9	8,529,950	73.6
ERR2775668	17,274,474	13,799,136	85.9	9,852,763	71.4
ERR2775669	16,786,939	13,525,029	86.3	9,802,873	72.5
ERR2775670	15,204,776	12,248,050	86.4	9,032,702	73.7
ERR2775671	11,461,974	9,100,759	84.9	6,509,055	71.5
ERR2775672	15,301,870	12,092,472	86.6	8,836,322	73.1
ERR2775673	12,247,350	9,678,899	86.2	6,823,627	70.5
ERR2775674	12,718,558	10,018,047	86.7	7,270,784	72.5
ERR2775675	13,279,182	10,362,507	86.9	7,643,066	73.7
ERR2775676	14,546,288	11,620,043	86.3	8,331,936	71.7
ERR2775677	17,174,540	13,807,456	86.3	9,910,284	71.8
ERR2775678	17,414,226	14,078,405	86.8	10,294,253	73.1
ERR2775679	17,246,132	13,643,475	86.4	9,756,837	71.5
ERR2775680	11,403,686	9,015,980	85.8	6,383,622	70.8
ERR2775681	15,452,512	12,419,639	86.5	8,957,969	72.1
ERR2775682	17,533,837	14,169,711	86.4	10,330,258	72.9
ERR2775683	13,905,859	11,119,814	86.6	8,117,696	73.0
ERR2775684	14,116,063	11,330,645	87.1	8,172,131	72.1
ERR2775685	10,588,464	8,189,077	86.3	5,835,669	71.3
ERR2775686	16,052,676	12,844,838	86.3	9,253,262	72.0
ERR2775687	11,974,411	9,290,605	85.9	6,610,776	71.2
ERR2775688	18,407,201	14,873,971	86.3	10,764,983	72.4
ERR2775689	13,247,963	10,477,587	86.3	7,505,028	71.6
ERR2775690	13,937,437	11,134,910	87.3	8,087,906	72.6

**Table 2 animals-13-01187-t002:** Differentially expressed genes (DEGs) were obtained from D1 vs. D3, D1 vs. D4, and D2 vs. D4 comparisons. For each DEG, the ENSEMBL identity number, the mean of the normalized counts, the log2 Fold Change (log2FC), the raw *p*-values, the adjusted *p*-values (padj), and gene symbol are reported.

**D1 vs. D3**						
**ENSEMBL ID**	**D1**	**D3**	**log2FC**	***p*-value**	**padj**	**Gene Symbol**
ENSSSCG00000001427	51.66	129.03	−1.44	1.19 × 10^−4^	0.034	*C4A*
ENSSSCG00000029886	14.61	32.98	−1.12	1.75 × 10^−4^	0.039	*LYVE1*
ENSSSCG00000012234	16.32	30.25	−1.09	2.03 × 10^−5^	0.014	*SRPX*
ENSSSCG00000004052	48.08	96.87	−1.07	3.30 × 10^−6^	0.012	*FNDC1*
ENSSSCG00000038162	31.58	59.87	−1.00	8.73 × 10^−5^	0.031	*CCL21*
ENSSSCG00000037706	12.59	26.32	−0.97	1.90 × 10^−5^	0.014	*PRKAR2B*
ENSSSCG00000018007	143.72	284.16	−0.97	4.30 × 10^−4^	0.064	*MYH3*
ENSSSCG00000011239	15.31	27.81	−0.88	5.14 × 10^−4^	0.071	*TRANK1*
ENSSSCG00000033919	53.23	91.72	−0.86	9.27 × 10^−4^	0.099	*DCLK1*
ENSSSCG00000040904	19.62	35.12	−0.86	1.39 × 10^−5^	0.014	*CLDN1*
ENSSSCG00000001570	235.46	402.93	−0.82	3.32 × 10^−4^	0.059	*PI16*
ENSSSCG00000007073	10.58	18.38	−0.81	5.45 × 10^−4^	0.074	*ISM1*
ENSSSCG00000037803	98.72	184.50	−0.80	6.47 × 10^−4^	0.080	*MARCKS*
ENSSSCG00000040337	49.27	83.66	−0.80	5.12 × 10^−4^	0.071	*AK4*
ENSSSCG00000029458	16.47	27.36	−0.80	3.95 × 10^−5^	0.019	*SLC16A2*
ENSSSCG00000037025	76.02	123.59	−0.79	3.41 × 10^−5^	0.017	*PLVAP*
ENSSSCG00000014088	24.90	43.19	−0.76	3.18 × 10^−4^	0.059	*IQGAP2*
ENSSSCG00000006082	86.84	152.88	−0.76	1.06 × 10^−4^	0.033	*MATN2*
ENSSSCG00000032015	12.79	21.14	−0.73	3.64 × 10^−4^	0.061	*SH3BGRL2*
ENSSSCG00000037292	23.96	13.65	0.74	7.61 × 10^−4^	0.089	*PLA2G4E*
ENSSSCG00000039921	18.60	9.49	0.97	3.98 × 10^−4^	0.064	*LOC100153854*
ENSSSCG00000044553	33.93	13.43	1.31	6.02 × 10^−4^	0.078	*DDIT3*
**D1 vs. D4**						
**ENSEMBL ID**	**D1**	**D4**	**log2FC**	***p*-Value**	**padj**	**Gene Symbol**
ENSSSCG00000003088	42.11	90.45	−1.03	2.46 × 10^−4^	0.077	*APOE*
ENSSSCG00000004052	48.08	89.79	−0.94	8.00 × 10^−5^	0.052	*FNDC1*
ENSSSCG00000005997	376.19	588.51	−0.71	1.59 × 10^−4^	0.065	*COL14A1*
ENSSSCG00000008991	147.17	270.34	−1.01	3.51 × 10^−5^	0.031	*FRAS1*
ENSSSCG00000010554	453.56	1182.12	−1.57	3.47 × 10^−5^	0.031	*SCD*
ENSSSCG00000014232	19.88	39.84	−1.14	2.53 × 10^−5^	0.031	*LOX*
ENSSSCG00000024149	44.81	80.94	−0.98	5.22 × 10^−6^	0.021	*ELOVL5*
ENSSSCG00000026383	41.57	63.66	−0.72	3.69 × 10^−5^	0.031	*NRP2*
ENSSSCG00000036236	81.34	160.13	−1.04	2.44 × 10^−4^	0.077	*ELOVL6*
ENSSSCG00000038420	54.49	96.94	−0.88	4.62 × 10^−5^	0.033	*PERP*
ENSSSCG00000040337	49.27	97.58	−0.94	9.07 × 10^−5^	0.053	*AK4*
ENSSSCG00000032450	38.68	18.34	0.99	7.54 × 10^−6^	0.021	*LYRM9*
**D2 vs. D4**						
**ENSEMBL ID**	**D2**	**D4**	**log2FC**	***p*-value**	**padj**	**Gene Symbol**
ENSSSCG00000016064	30.39	17.52	0.92	6.27 × 10^−6^	0.048	*TMEFF2*
ENSSSCG00000025053	121.60	187.28	−1.08	9.06 × 10^−6^	0.048	*RYBP*

**Table 3 animals-13-01187-t003:** Functions and pathways generated by functional analysis with Cytoscape were used for each comparison (D1 vs. D3 and D1 vs. D4) of the respective differentially expressed genes. For each category, there are numbers and symbols of genes of the category of belonging (categories of Gene Ontology (GO) or Reactome or KEGG Pathways); description; and False Discovery Rate (FDR). The pathways marked in bold are those of interest to the study.

**D1 vs. D3**				
N Genes	Category ^a^	Description	FDR ^b^	Gene symbol
2	GO MF	protein phosphatase activator activity	0.0039	*CALM3|CALM1*
6	GO MF	protein kinase binding	0.0044	*CD4|PRKAR2B|CALM3|CALM1|* *ACTB|MARCKS*
9	GO BP	negative regulation of molecular function	0.0108	*PRKAR2B|IQGAP2|CALM3|CALM1|ACTB|PI16|C4A|CD44|DDIT3*
5	GO BP	positive regulation of cytosolic calcium ion concentration	0.0108	*CD4|CCL21|CALM3|CALM1|DDIT3*
2	GO BP	hyaluronan catabolic process	0.0112	*LYVE1|CD44*
2	GO BP	regulation of cellular extravasation	0.0161	*PLVAP|CCL21*
7	GO BP	response to biotic stimulus	0.0176	*CD4|DCLK1|CCL21|CLDN1|C4A|* *CD44|DDIT3*
6	GO BP	cell adhesion	0.0176	*CD4|LYVE1|CCL21|CLDN1|SRPX|* *CD44*
3	GO MF	actin filament binding	0.0177	*IQGAP2|MYH3|MARCKS*
4	GO MF	enzyme inhibitor activity	0.0191	*PRKAR2B|IQGAP2|PI16|C4A*
5	GO MF	protein domain-specific binding	0.0226	*PRKAR2B|CALM3|CALM1|* *SH3BGRL2|DDIT3*
2	GO BP	release of sequestered calcium ion into cytosol	0.0255	*CCL21|DDIT3*
14	GO BP	localization	0.0336	*CD4|MATN2|DCLK1|CCL21|PRKAR2B|IQGAP2|CALM3|CLDN1|CALM1|ACTB|SRPX|CD44|DDIT3|SLC16A2*
2	GO BP	multicellular organismal water homeostasis	0.0348	*PRKAR2B|CLDN1*
3	GO BP	negative regulation of peptidase activity	0.0394	*PI16|C4A|CD44*
2	GO MF	virus receptor activity	0.0412	*CD4|CLDN1*
2	GO BP	positive regulation of actin filament polymerization	0.0432	*CCL21|IQGAP2*
2	GO BP	response to fatty acid	0.0436	*CCL21|CLDN1*
**5**	**GO BP**	**positive regulation of immune system process**	**0.0437**	** *CD4|PLVAP|CCL21|ACTB|C4A* **
5	GO BP	cell motility	0.0474	*MATN2|DCLK1|CCL21|ACTB|CD44*
2	GO BP	dendrite development	0.0474	*MATN2|DCLK1*
7	GO BP	small molecule metabolic process	0.0492	*LYVE1|PRKAR2B|CALM3|CALM1|* *AK4|CD44|MYH3*
**D1 vs. D4**				
N Genes	Category	Description	FDR	Gene symbol
**3**	**Reactome Pathways**	**Fatty acyl-CoA biosynthesis**	**0.0034**	** *SCD|ELOVL5|ELOVL6* **
**3**	**GO BP**	**Unsaturated fatty acid biosynthetic process**	**0.0146**	** *SCD|ELOVL5|ELOVL6* **
4	GO BP	Purine ribonucleotide biosynthetic process	0.0146	*SCD|ELOVL5|ELOVL6|AK4*
2	GO BP	Fatty acid elongation, saturated fatty acid	0.0146	*ELOVL5|ELOVL6*
2	GO BP	Fatty acid elongation, monounsaturated fatty acid	0.0146	*ELOVL5|ELOVL6*
2	GO BP	Fatty acid elongation, polyunsaturated fatty acid	0.0146	*ELOVL5|ELOVL6*
2	GO BP	Very long-chain fatty acid biosynthetic process	0.0146	*ELOVL5|ELOVL6*
3	GO BP	Regulation of cholesterol biosynthetic process	0.0146	*APOE|SCD|ELOVL6*
4	GO BP	Regulation of lipid biosynthetic process	0.0146	*APOE|SCD|ELOVL5|ELOVL6*
**3**	**GO BP**	**fatty-acyl-CoA biosynthetic process**	**0.0146**	** *SCD|ELOVL5|ELOVL6* **
5	GO BP	Regulation of small molecule metabolic process	0.0146	*APOE|SCD|ELOVL5|ELOVL6|AK4*
2	GO BP	Long-chain fatty-acyl-coa biosynthetic process	0.0215	*ELOVL5|ELOVL6*
2	GO MF	Fatty acid elongase activity	0.0400	*ELOVL5|ELOVL6*
2	GO MF	3-oxo-arachidoyl-CoA synthase activity	0.0400	*ELOVL5|ELOVL6*
2	GO MF	3-oxo-cerotoyl-CoA synthase activity	0.0400	*ELOVL5|ELOVL6*
2	GO MF	3-oxo-lignoceronyl-CoA synthase activity	0.0400	*ELOVL5|ELOVL6*
2	GO MF	Very-long-chain 3-ketoacyl-coa synthase activity	0.0400	*ELOVL5|ELOVL6*
2	KEGG Pathways	Biosynthesis of unsaturated fatty acids	0.0434	*SCD|ELOVL6*

^a^ MF = Molecular Function; BP = Biological Process. ^b^ FDR = False Discovery Rate.

**Table 4 animals-13-01187-t004:** Description of the functions of the most relevant differentially expressed genes (DEGs).

Comparisons	DEGs	Gene Function
D1 vs. D3	*C4A*	*C4A* (*complement C4A*) gene favors the reduction of susceptibility to infections as a deficiency of C4A and C4B proteins was associated with an increase in susceptibility to infections [23].
	*CCL21*	*CCL21* (*C-C motif chemokine ligand 21*) expresses proteins that are part of and promote immune cell migration processes. *CCL21* stimulates the migration of T cells and dendritic cells to specific regions of the node in secondary lymphoid organs, where antigen presentation can occur [24].
	*LYVE1*	*LYVE1* (*lymphatic vessel endothelial hyaluronan receptor 1*) expresses proteins that are part of and promote immune cell migration processes. *LYVE1* expresses a receptor that binds hyaluronic acid present on the membrane of dendritic cells, allowing passage of these cells through lymphatic vessels [25,26].
	*PLVAP*	*PLVAP* (*plasmalemma vesicle-associated protein*) expresses proteins that are part of and promote immune cell migration processes. *PLVAP* expresses a protein that acts as a physical filter for regulating the entry of lymphocytes and soluble antigens into the parenchyma [27].
D1 vs. D4	*ELOVL5*	*ELOVL5* (*ELOVL Fatty Acid Elongase 5*) is part of the enzymes group called Elongation of very-long-chain fatty acids (ELOVLs) that catalyze the elongation of two carbon atoms to polyunsaturated fatty acids (PUFAs). *ELOVL5* acts in the pathway that leads from alpha-linoleic acid, a polyunsaturated fatty acid of the omega-3 series and found in greater amounts in extruded linseed, to the synthesis of eicosapentaenoic acid (EPA) and docosahexaenoic acid (DHA) [28].
	*ELOVL6*	*ELOVL6* (*ELOVL Fatty Acid Elongase 6*) is part of the enzymes group called Elongation of very-long-chain fatty acids (ELOVLs) that catalyzes the elongation of two carbon atoms into saturated and monounsaturated fatty acids [29].
	*SCD*	*SCD* (*stearoyl-CoA desaturase*) is a key enzyme in unsaturated fatty acid biosynthesis, since it catalyzes the insertion of the first double bond into saturated fatty acyl-CoA substrates (palmitoyl-CoA and stearoyl-CoA) at the delta-9 position [30,31].

## Data Availability

The forty-eight RNA-seq datasets for pigs utilized for this paper were downloaded from the ArrayExpress, https://www.ebi.ac.uk/biostudies/arrayexpress, (accessed on 29 January 2023) accession: E-MTAB-7131 [2,4].

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
