# Peer review of "RNA-Seq Study on the Longissimus thoracis Muscle of Italian Large White Pigs Fed Extruded Linseed with or without Antioxidants and Polyphenols"

_animals, 2023, doi:10.3390/ani13071187_

Round 1

Reviewer 1 Report

The addition of the following antioxidants and polyphenols to the feed in pig production has been shown to have many favorable regulatory effects. In this study, n3-PUFA, vitamin E, selenium and plant extracts were added to pig diets alone or in combination, and changes in some gene pathways in muscle were explored through RNA-seq, which could provide some scientific basis for pig production. However, this paper has the following deficiencies:

1 The additives used in this study have been widely used in pig production and are not attractive to readers.

2. After the addition of these exogenous substances, some phenotypic characteristics of pigs were not measured, and it was impossible to explain which aspects of traits were affected by the changes in genes or signaling pathways caused by these substances.

3 The omics results of this experiment have not been verified by experiments. In general, transcriptomic data are validated by at least quantitative PCR.

4. The experimental design was that n3-PUFA in the diet was used in combination with vitamin E, selenium and plant extract, while lack of vitamin E, selenium and plant extract were used alone in the treatment group.

Author Response

Please find the point-by-point response in the attached file

Reviewer 2 Report

The whole work talks about an interesting topic and is clearly describes. It needs only a little correction like the explanation
 of the abbreviation of the name before first using them. Now, the explanation  is sometimes next to next using.

The publication is a continuation of similar research on gene expression in the Longissimus thoracic muscle. The obtained results complement previous publications on gene expression in the Large White Pig. Specific questions and concerns that authors can address are listed below:

From which large group of piglets was the research group selected?

From how many sows and boars were piglets born?

How were each diets e.g how much n-3 PUFAa were contents in total fat (D2) or how much polyphenols were in D4 ? How long were used all diets before slaughter ? Was the time used of each diet the same ? I didn't find information about storing Longissimus thoracis muscle before the research and about RNA extraction.

Wasn't studied the expression level of genes in the research ?

Why tables 2 and 3 do not contain all the results e.g. relating to D1sD2? The data should be in tabes or the authors should explain why tabes don't contain them.

The conclusions are consistent with the purpose of the study.

Author Response

(The authors gave the same response as above.)

Round 2

Reviewer 1 Report

The authors' answers do not change my opinion of the study. The study used antioxidant additives that other researchers have reported in pigs. Moreover, the experimental design of this study was not perfect, and only transcriptome sequencing data was insufficient to support the authors' conclusions.